# Deletion of the scavenger receptor *Scarb1* in osteoblast progenitors and myeloid cells does not affect bone mass

Michela Palmieri[1,2], Teenamol E. Joseph[1,2], Horacio Gomez-Acevedo[3], Ha-Neui Kim[1], Stavros C. Manolagas[1], Charles A. O'Brien[1,2,4], Elena Ambrogini[1,2,4]*

1 Department of Internal Medicine, Division of Endocrinology and Metabolism and Center for Musculoskeletal Disease Research, University of Arkansas for Medical Sciences, Little Rock, Arkansas, United States of America, 2 Department of Veterans Affairs, Central Arkansas Veterans Healthcare System, Little Rock, Arkansas, United States of America, 3 Department of Biomedical Informatics, University of Arkansas for Medical Sciences, Little Rock, Arkansas, United States of America, 4 Department of Orthopedic Surgery, University of Arkansas for Medical Sciences, Little Rock, Arkansas, United States of America

* eambrogini@uams.edu

## Abstract

The scavenger receptor class B member 1 (SCARB1), encoded by *Scarb1*, is a cell surface receptor for high density lipoproteins, low density lipoproteins (LDL), oxidized LDL (OxLDL), and phosphocholine-containing oxidized phospholipids (PC-OxPLs). *Scarb1* is expressed in multiple cell types, including osteoblasts and macrophages. PC-OxPLs, present on OxLDL and apoptotic cells, adversely affect bone metabolism. Overexpression of E06 IgM – a natural antibody that recognizes PC-OxPLs– increases cancellous and cortical bone at 6 months of age in both sexes and protects against age- and high fat diet- induced bone loss, by increasing bone formation. We have reported that SCARB1 is the most abundant scavenger receptor for OxPLs in osteoblastic cells, and osteoblasts derived from *Scarb1* knockout mice (*Scarb1* KO) are protected from the pro-apoptotic and anti-differentiating effects of OxLDL. Skeletal analysis of *Scarb1* KO mice produced contradictory results, with some studies reporting elevated bone mass and others reporting low bone mass. To clarify if *Scarb1* mediates the negative effects of PC-OxPLs in bone, we deleted it in osteoblast lineage cells using Osx1-Cre transgenic mice. Bone mineral density (BMD) measurements and micro-CT analysis of cancellous and cortical bone at 6 months of age did not reveal any differences between *Scarb1*[ΔOSX-I] mice and their wild-type (WT), Osx1-Cre, or *Scarb1*[fl/fl] littermate controls. We then investigated whether PC-OxPLs could exert their anti-osteogenic effects via activation of SCARB1 in myeloid cells by deleting *Scarb1* in LysM-Cre expressing cells. BMD measurements and micro-CT analysis at 6 months of age did not show any differences between *Scarb1*[ΔLysM] mice and their WT, LysM-Cre, or *Scarb1*[fl/fl] controls. Based on this

**Data availability statement:** All relevant data are within the paper and its Supporting Information files.

**Funding:** This work was supported by the Biomedical Laboratory Research and Development Service of the Veterans Administration Office of Research and Development (1I01BX003901 to EA), the National Institutes of Health (P20 GM125503 to CAO), the University of Arkansas for Medical Sciences Tobacco Funds and Translational Research Institute (239 G1-50893-01; 1UL1 RR-029884 to EA). The funders had no role in the design of the study, in the collection, analyses, or interpretation of the data, in the writing of the manuscript or in the decision to publish the results.

**Competing interests:** The authors have declared that no competing interests exist.

evidence, we conclude that the adverse skeletal effects of PC-OxPLs in adult mice are not mediated by *Scarb1* expressed in osteoblast lineage cells or myeloid cells.

## Introduction

The scavenger receptor class B member 1 (SCARB1), encoded by *Scarb1* in mice, is a glycosylated cell surface receptor for high density lipoproteins (HDL), most abundantly expressed in the liver and in steroidogenic tissues such as adrenal glands, ovaries, and testis [1]. SCARB1 is responsible for the uptake of cholesteryl esters from HDL, as well as efflux of cellular cholesterol to HDL [2,3]. In the liver, clearance of HDL cholesteryl esters is essential for anti-atherogenic reverse cholesterol transport [4] and bile acid production, whereas in the adrenal glands it is essential for optimal glucocorticoid generation [5]. Consistent with this, patients with heterozygous loss-of-function mutation of *SCARB1* have increased risk of adrenal insufficiency [6] and cardiovascular diseases, despite a significant increase in HDL-cholesterol concentration in the serum [7,8]. *SCARB1* is expressed in many other tissues and cell types including adipocytes, endothelial and epithelial cells, monocytes/macrophages, osteoblasts [9], and has been found to be important in many other processes such as regulation of platelet physiology [5] inflammation [7], regulation of bacterial invasion into cells [10], and cancer growth and metastasis [7].

In addition to HDL, SCARB1 can bind, albeit with lower affinity, to low-density lipoproteins (LDL). In endothelial cells, SCARB1 mediates LDL transcytosis and promotes atherosclerosis [11]. Moreover, SCARB1 is a receptor for very-low-density lipoproteins (VLDL), lipoprotein(a) – Lp(a)-, which is primary carrier of oxidized phospholipids [1,12,13], oxidized low-density lipoproteins (OxLDL) and phosphocholine-containing oxidized phospholipids (PC-OxPLs) [1,14,15]. In osteoblasts, SCARB1 has been implicated in the uptake of OxLDL, cholesteryl ester, and estradiol [12,16].

We have previously shown that physiological levels of PC-OxPLs reduce bone mass. Specifically, we found that both male and female transgenic mice expressing a single-chain variable fragment (scFv) of the antigen-binding domain of the natural E06 IgM antibody (E06-scFv), which binds and neutralizes PC-OxPLs, exhibited increased cancellous and cortical bone mass at 6 months of age [9], and were protected from the bone loss caused by a high fat diet or aging [17,18]. These mice display increased osteoblast number and bone formation rate at both cancellous and cortical sites, reduced osteoblast apoptosis *in vivo*, and decreased osteoclast number in vertebral bone throughout life [9,17,18].

In addition to SCARB1, PC-OxPL is recognized by the scavenger receptor CD36 and by the toll-like receptors 2, 4 and 6 [14], all of which are expressed in osteoblastic cells [12,19,20], as well as in osteoclasts and bone marrow macrophages [14,21]. We have previously shown that *Scarb1* is the most abundant scavenger receptor for PC-OxPLs in calvaria-derived osteoblastic cells, as determined by qPCR, and silencing *Scarb1* protects calvaria-derived osteoblastic cells from OxLDL-induced apoptosis

[9]. Importantly, our studies indicated that both marrow-derived and calvaria-derived osteoblasts from mice with germline deletion of *Scarb1* (*Scarb1* KO mice) are protected from the pro-apoptotic and anti-differentiating effects of OxLDL [9]. Some studies, but not all, have shown that *Scarb1* KO mice have increased osteoblast number, bone formation rate, and high bone mass as well as histomorphometric similarities with E06-scFv transgenic mice [19,22,23]. However, the mechanisms by which PC-OxPLs affect bone mass remain unclear. It is unknown if PC-OxPLs exert their deleterious effects on osteoblasts directly, by binding to scavenger receptors such as SCARB1, or indirectly by acting on other cells.

We hypothesized that SCARB1 is an essential mediator of the pro-apoptotic effects of PC-OxPLs on osteoblasts and that E06-scFv prevents the binding of PC-OxPLs to SCARB1 on these cells thus reducing the pro-inflammatory actions of OxPLs [14]. Therefore, we postulated that deleting *Scarb1* in cells of the osteoblast lineage would increase osteoblastogenesis and recapitulate the bone phenotype seen in both female and male E06-scFv transgenic mice at 6 months of age, fed a chow diet [9]. To this end, we deleted *Scarb1* in cells of the osteoblast lineage using an Osx1-Cre transgene and analyzed the bone phenotype in adult female and male mice. We show that deletion of *Scarb1* using Osx1-Cre mice did not affect bone mass or architecture, suggesting that SCARB1 is dispensable for osteoblast differentiation and function and does not mediate the deleterious effects of PC-OxPLs on osteoblasts or bone.

We then tested the alternative hypothesis that PC-OxPLs may exert their anti-osteogenic effects via activation of SCARB1 in macrophages, possibly leading to increased production of anti-osteoblastogenic cytokines, such as TNF-α [24]. To this end, we deleted *Scarb1* specifically in myeloid cells using LysM-Cre mice. Lack of *Scarb1* in myeloid progenitors and their descendants also did not alter bone mass *in vivo*.

We conclude that *Scarb1* expression in either osteoblasts or osteoclasts is not involved in bone homeostasis and, therefore, *Scarb1* expressed by these cells is not a major mediator of the deleterious effects of PC-OxPLs on bone.

## Materials and methods

### Animals

C57BL/6J (stock number 000664) and *Scarb1* KO mice (stock number 003379) were obtained from the Jackson Laboratories. The mouse line harboring the *Scarb1* conditional allele was kindly provided by Philip Shaul, University of Texas Southwestern Medical Center, Dallas, TX [11]. To delete *Scarb1* in the entire osteoblast lineage, we crossed *Scarb1* floxed mice with Osx1-Cre transgenic mice obtained from the Jackson laboratories (stock number 006361) [25]. The Osx1-Cre transgene becomes active at the earliest stages of osteoblast differentiation [25] and lineage-tracing studies have established that all osteoblasts and osteocytes are derived from progenitors labeled by the Osx1-Cre transgene [26]. We have used this strain extensively for conditional gene inactivation in osteoblast-lineage cells [26–36].

We used a two-step breeding strategy to obtain experimental animals. To delete *Scarb1* in the osteoblast lineage we initially crossed hemizygous Osx1-Cre transgenic mice with homozygous *Scarb1* floxed mice to generate heterozygous *Scarb1* floxed mice with and without a Cre allele. Those mice were used in a second cross to generate the three control groups and the experimental mice: WT, Osx1-Cre, mice homozygous for the *Scarb1*-floxed allele, hereafter referred to as *Scarb1*^fl/fl, and the experimental mice Osx1-Cre; *Scarb1*^fl/fl hereafter referred as *Scarb1*^ΔOSX1). All mice from the first cross were fed a doxycycline-containing diet (BioServ doxycycline diet (S3888) 200 mg/kg (Bio Serv, Flemington, NJ, USA). Mice of the second cross were fed a doxycycline-containing diet from conception until weaning, at which time they were switched to regular chow [LabDiet 5K67 Mouse/Auto6F Diet (LabDiet, St. Louis, MO, USA)] to activate the Cre transgene.

To delete *Scarb1* in the entire myeloid lineage, *Scarb1*^fl/fl were crossed with LysM-Cre mice, purchased from Jackson Laboratories (stock number 004781) [37]. All mice were in the C57BL/6 background. We used the same two-step breeding strategy described above to obtain mice lacking *Scarb1* in the myeloid lineage. This generated the three control groups: WT, LysM-Cre, and *Scarb1*^fl/fl and the experimental mice: LysM-Cre; *Scarb1*^fl/fl, hereafter referred as *Scarb1*^ΔLysM). All mice were fed regular chow diet (LabDiet 5K67 Mouse/Auto6F Diet).

Mice were group housed under specific pathogen-free conditions and maintained at a constant temperature of 23°C, in a 12:12-hour light-dark cycle; they had ad libitum access to diet and water.

We genotyped the offspring of both lines by PCR using the following primer sequences: Cre-UP2: 5'-GCTAAACAT GCTTCATCGTCGG-3', Cre-DN2: 5'-GATCTCCGGTATTGAAACTCCAGC-3', product size 650 bp; *Scarb1*-Fwd: 5'-GCACAGAGGACCCAACAGCGCACAAAATGG-3', *Scarb1*-Rev: 5'-GCTGGGATTCAAGGTGTGTGCCACCACTAC-3', primer for detection of Cre recombination: 5'-AGACCAATGGACCCTGTGCTTGGAGTGAGC-3', product size wild type 149 bp; floxed allele before recombination 188 bp, floxed allele after recombination 315 bp.

## Imaging

Bone mineral density (BMD) measurements and percentages of lean and fat body mass were calculated by dual-energy X-ray absorptiometry (DXA) of sedated mice (2% isoflurane) using a PIXImus densitometer (GE Lunar) [9,38]. The mean coefficient of variation, calculated using a proprietary phantom scanned at the beginning of each session for the mice used in this study, was 0.38% for BMD and 0.15% for the percentage of body fat.

Bone microarchitecture was measured using VivaCT 80 scanner (Scanco Medical AG, Bruettisellen, Switzerland). We measured cancellous bone at the fifth lumbar vertebra (L5) and left femur and the cortical bone at the left femoral diaphysis (midpoint of the bone length as determined at scout view). Briefly, bones were dissected and cleaned from soft tissues. L5 and the left femur were fixed using 10% Millonig's Neutral Buffered Formalin (Leica Biosystems Inc., Buffalo Grove, IL, USA) with 0.5% sucrose. After fixation, bones were dehydrated in solutions containing progressively increasing ethanol concentrations and kept in 100% ethanol until analysis. For the scan, vertebrae and femora were loaded into scanning tubes and scanned at 10um nominal isotropic voxel size, 500 projections (FOV/Diameter 31.9 mm E = 70 kVp, I = 114 µA, 8W, Integration time 200ms and threshold 200 mg/cm$^3$), and integrated into 3-D voxel images (1024x1024-pixel matrices for each individual planar stack). During the conduct of these studies, the mean coefficient of variation of the micro-CT phantom was performed weekly and was 0.16%. The entire vertebral body was scanned to obtain a number of slices ranging between 310 and 330. Femora were scanned from the distal epiphysis to the mid-diaphysis to obtain a number of slices ranging between 750 and 810. For the analysis, two-dimensional evaluation of cancellous bones was performed on contours of the cross sectional acquired images; primary spongiosa and cortex were excluded. On the vertebral body, contours were drawn from the rostral to the caudal growth plate to obtain 220–250 slices (10µm/slice) and bone outside the vertebral body plate was excluded. The evaluation of the cancellous bone at the distal femur was performed on contours drawn from the distal metaphysis to the diaphysis to obtain 151 slices (10 µm/slice). All cancellous bone measurements were made by drawing contours every 10 to 20 slices; voxel counting was used for bone volume per tissue volume measurements and sphere filling distance transformation indices were used for cancellous microarchitecture with a threshold value of 285, without pre-assumptions about the bone shape as a rod or plate.

Two-dimensional evaluation of cortical bone in femur was performed at mid-diaphysis. Contours were drawn at mid-diaphysis to obtain 40 slices (10µm/slice) with a threshold unit of 260 for cortical thickness. A Gaussian filter (sigma = 1.2, sigma = 0.8, support = 1) was applied to all analyzed scans (cancellous and cortical bone respectively) to reduce signal noise.

## Culture of osteoblastic cells

Calvaria cells were isolated from C57BL/6J and *Scarb1* KO littermate neonatal mice by sequential digestion with collagenase type 2 (Worthington, Columbus OH, USA, cat. CLS-2, lot 47E17554B) as previously described [39]. The cells were cultured in α-MEM medium (Invitrogen, Carlsbad, CA, USA, cat. 11900–0.24) containing 10% Premium Select fetal bovine serum (FBS) (Atlanta Biologicals, Flowery Branch, GA, USA), 1% penicillin/streptomycin/glutamine (PSG) and L-Ascorbic Acid Phosphate (Wako, Richmond, VA, USA cat. 013–12061) for twenty-one days. Gene expression was quantified by PCR as indicated below. Bone marrow cells were obtained by flushing cells from the femoral diaphysis (after removing

the proximal and distal ends) and culturing them with α-MEM medium containing 10% Premium FBS, 1% PSG and 1 mM L-Ascorbic Acid Phosphate up to 80% confluence. Proliferation was measured by BrdU incorporation with the Cell Proliferation ELISA kit from Roche Diagnostics (Roche Diagnostics, Indianapolis, IN, USA) following the manufacturer's instructions. Triplicate cultures were analyzed for all assays. Oxidized LDL (OxLDL) was obtained from Alfa Aesar (Alpha Aesar, Haverhill, MA, USA).

## RNA isolation, cDNA synthesis, and Real time Quantitative PCR (RT-qPCR)

Total RNA was extracted from calvaria cells with Trizol (Thermo Fisher Scientific, Waltham, MA, USA) and purified with Direct-zol RNA Miniprep (Zymo Research, Irvine, CA, USA cat. R2050) according to the manufacturer's instructions. RNA was then quantified using a NanoDrop instrument (Thermo Fisher Scientific), and its integrity was verified by resolution on 0.8% agarose gels. Complementary DNA (cDNA) was reverse transcribed from 0.5 µg of total RNA extract using the High-Capacity cDNA Reverse Transcription kit (Applied Biosystems, Foster City, CA, USA cat. 4368813) according to manufacturer's instructions. PCR was performed using TaqMan Gene Expression Assays manufactured by Applied Biosystems, as listed in Supporting information S1 Table. Transcript levels were calculated by normalizing to the reference mRNA Mitochondrial Ribosomal Protein S2 (*Mrps2*) using the ΔCt method [40]. We used *Mrps2* as a housekeeping gene because it encodes a mitochondrial ribosomal protein which displays little or no change in expression under a variety of conditions.

## Genomic DNA isolation and Taqman assay to quantify gene deletion

To quantify the gene deletion in *Scarb1*$^{\Delta OSX1}$ mice, we obtained genomic DNA from cortical bone of Osx1-Cre and *Scarb1*$^{\Delta OSX1}$. Briefly, after dissection, the distal and proximal ends of femur and tibia were removed, and bone marrow cells were flushed out from the bone cavity with PBS. The surfaces of the bone shafts were scraped with a scalpel to remove the periosteum. Femoral and tibial cortical bones were placed in 14% EDTA solution for 8 days to allow decalcification. After decalcification, the bones were washed twice with water to eliminate EDTA, cut into 2–3 pieces, and placed in an Eppendorf tube to proceed with genomic DNA digestion and purification. Decalcified bone was digested with proteinase K (Cat. P5850, Sigma-Aldrich, St. Louis MO, USA) (0.67 mg/ml proteinase K in 30mM TRIS pH 8.0, 200mM NaCl, 10mM EDTA, and 1% SDS) at 55° C overnight. Genomic DNA was isolated by phenol/chloroform extraction and ethanol precipitation. Spleen was used as negative control and was harvested, cut into 3–4 pieces, and immediately frozen in liquid nitrogen. For genomic DNA extraction, spleen fragments were digested with proteinase K (0.67 mg/ml in 30mM TRIS pH 8.0, 200mM NaCl, 10mM EDTA, and 1% SDS) at 55° C overnight and genomic DNA was isolated by phenol/chloroform extraction and ethanol precipitation.

To quantify gene deletion in *Scarb1*$^{\Delta LysM}$ mice, we obtained genomic DNA from macrophages of LysM-Cre and *Scarb1*$^{\Delta LysM}$. Briefly bone marrow cells were obtained by flushing femur and tibia and cultured in a-MEM with 10% FBS in the presence of 10ng/ml of M-CFS (R&D Systems rhM-CSF cat# 216-MC) for 24 hours. Non-adherent cells (myeloid lineage) were expanded for 3–4 days in the presence of 10ng/ml of M-CSF. At the end of the culture genomic DNA was extracted using the Qiagen QIAamp DNA mini kit (Cat. 51304, Qiagen, Germantown, MD, USA). Spleen was used as negative control as indicated above.

A custom TaqMan assay was obtained from Applied Biosystems to quantify the *Scarb1* deletion efficiency in genomic DNA of both strains: Fwd 5'- GGACTGTGTGTGGGTGTGT'; Rev 5'- TTCTGTCTCTGGAGCAATCAATCTC-3'; probe 5'-CTGCCATGCTGAGTTTT-3'. The relative amount of genomic DNA was calculated with the ΔCt method using an assay for the transferrin receptor gene (Tfrc) as a control, as listed in Supporting information S1 Table [40].

## Statistics

No experimentally derived data were excluded. One of the female Osx1-Cre mice was not harvested with the rest of the mice because it died soon after the DXA measurement at 6 months of age. The micro-CT measurement was not

performed in this mouse and therefore, the number of female mice in the Osx1-Cre group was 14 for Fig 2A-2C and in panels A-C in S2 Fig and 13 in Fig 2D-2F, S3 Fig and panels A-C in S5 Fig. In panel F of S5 Fig, the femoral length could not be measured in 1 male *Scarb1*<sup>fl/fl</sup> mouse because the head of the femur was damaged during the harvest.

Each legend includes the number of mice or samples used in each experiment. Single data points are shown in Figs 1, 2D-2F, 3D-3F, 4D-4F, 5D-5F and S1, S3-S6, S8-S10 Figs with mean ± standard deviation. In Figs 2A-2C, 3A-3C, 4A-4C, 5A-5C and S2 and S7 Figs, the data are shown as mean ± standard deviation.

Statistical analyses were performed using GraphPad Prism (version 10). Group mean values were compared by Student's two-tailed *t*-test or ANOVA as appropriate. When ANOVA indicated a significant effect, pairwise multiple comparisons were performed and the p-values adjusted using the Tukey's pairwise comparison procedure or the Holm-Sidak method as appropriate. Statistical analysis for the data shown in Figs 2A-2C, 3A-3C, 4A-4C, 5A-5C and S2 and S7 Figs, was performed using ANOVA repeated measures by R (version 4.5). The p-value reported as not significant is > 0.05.

For the *in vivo* studies, the sample size was adequate to detect a difference of 1.2 standard deviations at a power of 0.8, and p < 0.05 [24]. For *in vitro* experiments, the number of replicates was sufficient to provide confidence in the measurements.

All data were collected and analyzed by personnel blinded to the identity of the samples. The Raw data are reported in the Supporting information S1 File.

## Ethics

This study was carried out in strict accordance with the recommendations in the Guide for the Care and Use of Laboratory Animals of the National Institutes of Health. The animal protocols were approved by the Institutional Animal Care and Use Committees of the University of Arkansas for Medical Sciences (Animal Use Protocol #3809) and the Central Arkansas Veterans Healthcare System (IACUC protocol # 1400199). Anesthesia was provided by inhalation of 2% Isofluorane. Euthanasia was performed by $CO_2$ inhalation from a compressed gas tank at a displacement rate of 10% to 30% volume/minute until all movement ceased followed by an additional 1 minute in the chamber. Death was verified by lack of respiration and cervical dislocation or decapitation.

## Results

### Global deletion of *Scarb1* increases osteoblast differentiation and protects against the anti-proliferative effects of OxLDL

The result of our published work strongly suggested that *Scarb1* could be mediating the deleterious effect of oxidized phospholipids on osteoblastic cells in vitro [9] Herein, we further expand those studies and show that calvaria-derived osteoblasts obtained from *Scarb1* KO mice, cultured for 21 days, expressed more osteocalcin and alkaline phosphatase compared to cells from WT mice (**Fig 1A**). Moreover, bone marrow-derived osteoblasts obtained from *Scarb1* KO mice were protected against the decreased proliferation caused by OxLDL (**Fig 1B**). These results, together with previous published work [9] suggested that *Scarb1* may mediate the effects of OxLDL in osteoblasts by affecting osteoblast proliferation, in addition to apoptosis and differentiation.

### Deletion of *Scarb1* in Osx1-Cre-targeted cells does not affect body weight and fat mass

To investigate the role of *Scarb1* in osteoblasts we deleted this gene in cells targeted by the Osx1-Cre transgene. We crossed *Scarb1*<sup>fl/fl</sup> mice with transgenic mice expressing Cre recombinase under the control of Osx1 (Osx1-Cre mice) regulatory elements [25]. The *Scarb1*<sup>fl/fl</sup> mice, harboring the *Scarb1* allele with loxP sites inserted in intron 1 and intron 3 [11], were provided by Philip Shaul at UT Southwestern. The phenotype of the experimental *Scarb1*<sup>ΔOSX1</sup> mice was compared with WT, Osx1-Cre and *Scarb1*<sup>fl/fl</sup> littermate controls.

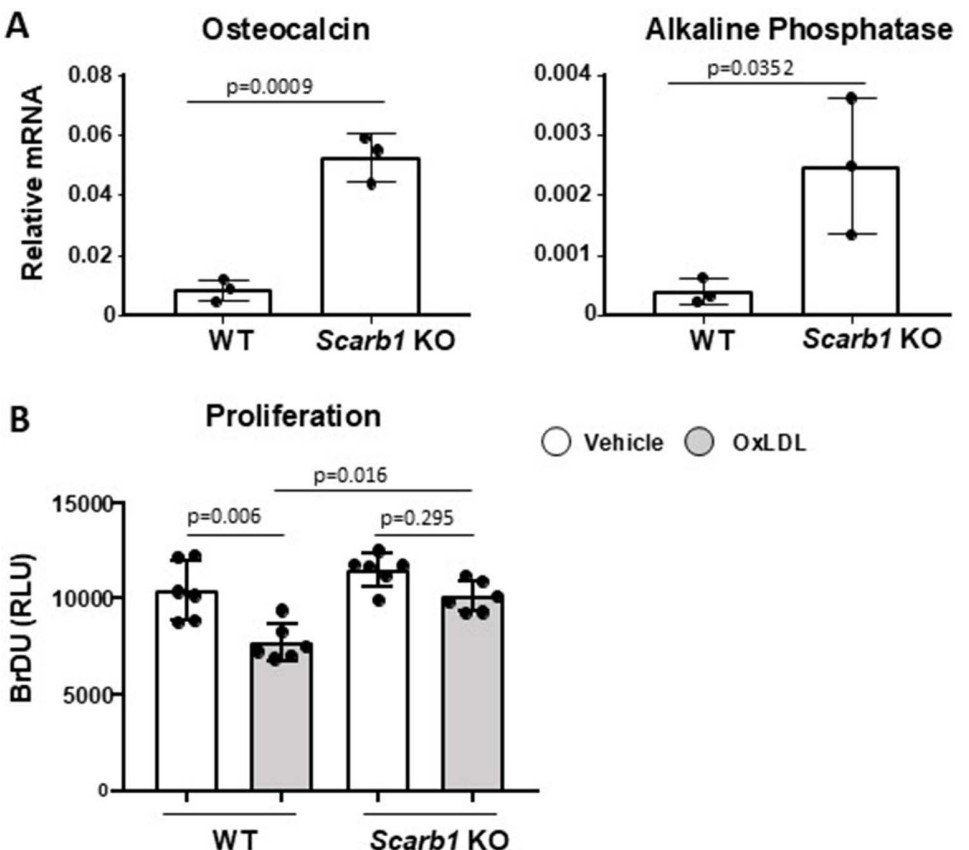

**Fig 1. Deletion of *Scarb1* increases osteoblast differentiation and protects against the anti-proliferative effects of OxLDL.** (A) The expression of osteocalcin and alkaline phosphatase was quantified with RT-PCR in calvaria-derived osteoblasts from newborn C57BL6/J (n = 5) and *Scarb1* KO mice (n = 2) cultured for 21 days. Transcripts were normalized to the housekeeping gene *Mrps2*. Data analyzed by Student's t-test. (B) Proliferation was measured with Bromodeoxyuridine (BrDU) incorporation in bone marrow-derived osteoblasts from 4-5-month-old WT and *Scarb1* KO mice (n = 3/group) 3 days after direct addition to the cultures of vehicle or OxLDL (50 μg/ml). Data analyzed by ANOVA; p-values were adjusted using the Holm-Sidak multiple comparison procedure. Data are shown as individual values with mean and standard deviation. All measures were performed using three or six technical replicates. RLU, relative light units. OxLDL, oxidized low density lipoproteins.

We first quantified the deletion of the *Scarb1* gene in cortical bone and found that the levels of *Scarb1* floxed exons were 45.5% and 24.3% lower in the cortical bone of femur and tibia of 6-month-old female and male *Scarb1*$^{\Delta OSX1}$, respectively, as compared with Osx1-Cre littermate controls, confirming deletion in bone (S1 Fig). This level of deletion is comparable to that obtained in other experiments where we used Osx1-Cre mice to delete other genes in the osteoblast lineage [26–28,34]. There was no change of *Scarb1* levels in the spleen, confirming the specificity of the deletion (S1 Fig).

The Osx1-Cre transgene alone causes mild growth plate/cranial defects during early development. However, most of these effects normalize with age [41,42]. All mice gained weight throughout the observational period and did not present any phenotypic differences.

Female mice carrying the Osx1-Cre transgene (Osx1-Cre and *Scarb1*$^{\Delta OSX1}$) had lower weight compared to WT and *Scarb1*$^{fl/fl}$ mice (p = 0.0005) and gained less weight with time (p = 0.004). Mice carrying the *Scarb1* floxed gene (*Scarb1*$^{fl/fl}$ and *Scarb1*$^{\Delta OSX1}$) had higher weight (p = 0.047) but gained similar weight with time compared to WT and Osx1-Cre combined (**panel A in** S2 Fig and S3 Table). In addition, mice carrying the Osx1-Cre transgene (Osx1-Cre and *Scarb1*$^{\Delta OSX1}$) gained less fat mass and lost less lean mass compared to WT and *Scarb1*$^{fl/fl}$ mice combined (p = 0.030) (**panels B-C in**

). In males, there was no difference in weight between the four groups (**panels D in** S2 Fig). However, male mice carrying the Osx1-Cre transgene (Osx1-Cre and *Scarb1*^ΔOSX1^) had higher fat mass and lower lean mass compared to WT and *Scarb1*^fl/fl^ mice (p = 0.004) (**panels E-F in** S2 Fig **and** S3 Table).

We could not detect in either sex and at any time point, any difference between *Scarb1*^ΔOSX1^ and the other three groups, indicating that, overall, deletion of *Scarb1* in osteoblast progenitors does not affect weight, fat mass or lean mass.

## Deletion of *Scarb1* using Osx1-Cre does not affect bone mass

The bone phenotype was analyzed by BMD by DXA at 2 and 6 months of age, and micro-CT of vertebral and femoral cancellous bone, as well as femoral cortical bone, at 6-months of age.

In females, there were no differences in spinal BMD at baseline between the four groups. However, the *Scarb1*^ΔOSX1^ mice gained less spinal BMD with time compared to the control groups (p interaction 0.005) (**Fig 2A and** S2 Table). At the femur, we found that the mice carrying the Osx1 transgene (Osx1-Cre and *Scarb1*^ΔOSX1^) had lower BMD compared to WT and *Scarb1*^fl/fl^ combined (p = 0.0007) but there were no changes in BMD with time between the 4 groups (**Fig 2B** and S2 Table). Similarly to the femur, the mice carrying the Osx1 transgene (Osx1-Cre and *Scarb1*^ΔOSX1^) had lower total BMD compared to WT and *Scarb1*^fl/fl^ mice (p = 0.0012) (**Fig 2C** and S2 Table). In addition, mice carrying the *Scarb1* floxed allele (*Scarb1*^fl/fl^ and *Scarb1*^ΔOSX1^) gained less total BMD with time compared to WT and Osx1-Cre mice (p = 0.007) (**Fig 2C** and S2 Table). However, with the exception of the changes in BMD at the spine, we did not detect any difference in BMD between *Scarb1*^ΔOSX1^ mice and the other 3 groups, indicating that, deletion of *Scarb1* in Osx1-Cre expressing cells does not affect bone mass.

The decrease in BMD at the spine in *Scarb1*^ΔOSX1^ was not confirmed by micro-CT analysis. In fact, micro-CT measurements showed an increase in bone volume/total volume (BV/TV) of the vertebra in the Osx1-Cre mice and *Scarb1*^ΔOSX1^ mice compared to WT, but there was no difference between Osx1-Cre and *Scarb1*^ΔOSX1^ mice, indicating that the increase in cancellous bone was due to the Osx1-Cre transgene and not to the deletion of *Scarb1* (Fig 2D). Similarly, in the cancellous bone of the femur there was an increase in BV/TV in the Osx1-Cre mice compared to WT and *Scarb1*^fl/fl^ and in *Scarb1*^ΔOSX1^ mice compared to WT mice, but there was no difference between Osx1 and *Scarb1*^ΔOSX1^ mice (Fig 2E).

Analysis of the cancellous bone microarchitecture revealed that, in vertebrae, there was an increase in trabecular number and trabecular thickness with decreased trabecular separation in Osx1-Cre mice compared to WT mice (**panels A-C in** S3 Fig). In the femur there was an increased trabecular number and decreased trabecular separation in the Osx1-Cre mice compared to WT and *Scarb1*^fl/fl^ and in *Scarb1*^ΔOSX1^ mice compared to WT mice and *Scarb1*^fl/fl^, but there was no difference between Osx1-Cre and *Scarb1*^ΔOSX1^ mice (**panels D-F in** S3 Fig). The increase in trabecular number was previously reported in 6-week-old Osx1-Cre mice only when this parameter was corrected by body weight [41].

We have reported previously that Osx1-Cre mice exhibit decreased femoral cortical thickness due to decreased periosteal apposition [27,30]. In females, we observed decreased cortical thickness in the Osx1-Cre mice and in *Scarb1*^ΔOSX1^ mice compared to WT and *Scarb1*^fl/fl^ but, in analogy to the cancellous bone, but there was no difference between Osx1-Cre and *Scarb1*^ΔOSX1^ mice (Fig 2F). Total area was decreased in Osx1-Cre and *Scarb1*^ΔOSX1^ mice compared to *Scarb1*^fl/fl^ (**panel A in** S5 Fig), and there was no change in medullary area between the four groups (**panel B in** S5 Fig). Femoral length was increased in *Scarb1*^fl/fl^ compared to WT mice and decreased in Osx1-Cre and *Scarb1*^ΔOSX1^ compared to WT mice (**panel C in** S5 Fig). These results indicate that deletion of *Scarb1* in osteoblast progenitors does not affect bone mass in female mice.

In male mice, there were no differences in spinal BMD between the four groups. (**Fig 3A**). Similarly to females, the mice carrying the Osx1 transgene (Osx1-Cre and *Scarb1*^ΔOSX1^) had lower femoral and total BMD compared to WT and *Scarb1*^fl/fl^ (p = 0.043 and p = 0.037 respectively) (**Fig 3B,3C and** S2 Table). In addition, mice carrying the *Scarb1* floxed allele (*Scarb1*^fl/fl^ and *Scarb1*^ΔOSX1^) gained more femoral BMD with time compared to WT and Osx1-Cre mice (p = 0.046) (**Fig 3B and** S2 Table). We could not detect any change in BMD in *Scarb1*^ΔOSX1^ compared to the other 3 groups of controls.

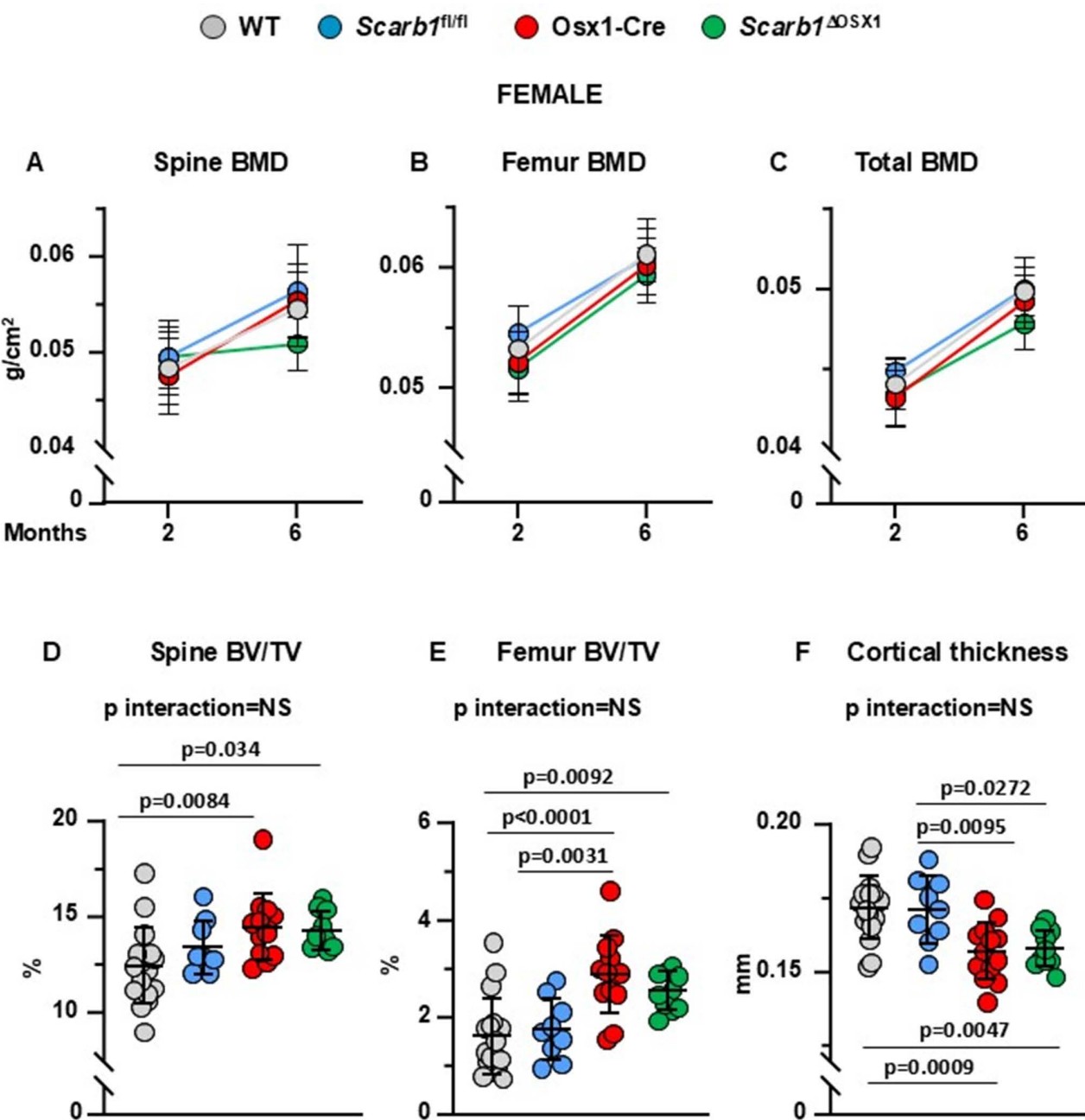

**Fig 2. Deletion of *Scarb1* in Osx1-Cre expressing cells does not affect bone mass in females.** (A-C) Determinations of spinal, femoral and total BMD by DXA in 2- and 6-month-old mice [WT n = 17; *Scarb1*^fl/fl n = 9; Osx1-Cre n = 14; *Scarb1*^ΔOSX1 n=10] Data are shown as mean and standard deviation. Adjusted p-values <0.05, calculated by repeated measures using two-way ANOVA, are shown in S2 Table. BMD: bone mineral density. (D-F) Micro-CT analysis of cancellous bone in vertebra and femoral metaphysis, and cortical bone at midshaft in 6-month-old mice [WT n = 17; *Scarb1*^fl/fl n = 9; Osx1-Cre n = 13; *Scarb1*^ΔOSX1 n = 10]. Data analyzed by 2-way ANOVA; the p-values were adjusted using the Tukey's pairwise comparison procedure. BV/TV, bone volume/total volume. NS: not significant, p > 0.05.

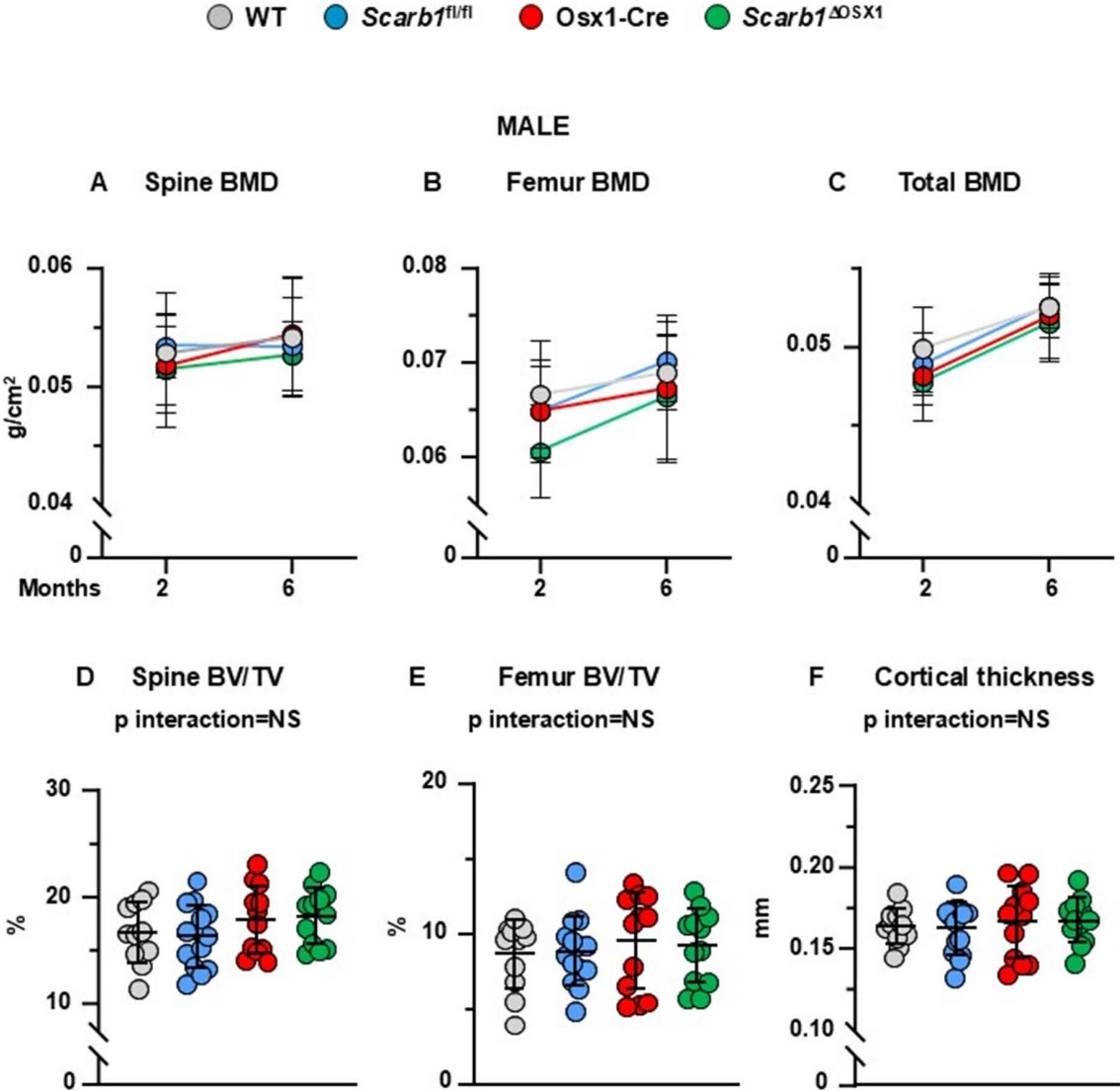

**Fig 3. Deletion of *Scarb1* in Osx1-Cre expressing cells does not affect bone mass in males.** (A-C) Determinations of spinal, femoral and total BMD by DXA in 2- and 6-month-old mice [WT n = 11; *Scarb1*<sup>fl/fl</sup> n = 14; Osx1-Cre n = 12; *Scarb1*<sup>ΔOSX1</sup> n = 12]. Data are shown as mean and standard deviation. Adjusted p-values <0.05, calculated by repeated measures using two-way ANOVA, are shown in S2 Table. BMD: bone mineral density. (D-F) Micro-CT analysis of cancellous bone in vertebra and femoral metaphysis, and cortical bone at midshaft in 6-month-old mice [WT n = 11; *Scarb1*<sup>fl/fl</sup> n = 14; Osx1-Cre n = 12; *Scarb1*<sup>ΔOSX1</sup>n = 12]. Data analyzed by 2-way ANOVA; the p-values were adjusted using the Tukey's pairwise comparison procedure. BV/TV, bone volume/total volume.. NS: not significant, p > 0.05.

The micro-CT analysis did not show any difference in the cancellous bone in either the vertebra or the femur between the four groups (Fig 3D,**3E**). Analysis of the cancellous bone microarchitecture revealed that, similar to femoral cancellous bone in female, there was an increased trabecular number and decreased trabecular separation in both vertebrae and femur in the Osx1-Cre mice compared to WT and *Scarb1*<sup>fl/fl</sup> and in *Scarb1*<sup>ΔOSX1</sup> mice compared to WT and *Scarb1*<sup>fl/fl</sup> mice, but there was no difference between Osx1-Cre and *Scarb1*<sup>ΔOSX1</sup> mice (S4 Fig).

In addition, we did not detect any changes in femoral cortical thickness (Fig 3F) or femoral length **(panel F in** S5 Fig**).** Total area was decreased in *Scarb1*<sup>ΔOSX1</sup> mice compared to *Scarb1*<sup>fl/fl</sup> mice **(panel D in** S5 Fig**)** and medullary area was decreased in Osx1-Cre and *Scarb1*<sup>ΔOSX1</sup>mice compared to *Scarb1*<sup>fl/fl</sup> mice **(panel E in** S5 Fig**),**

Overall, these results indicate that deletion of *Scarb1* in cells of the osteoblast lineage does not alter BMD, cancellous and cortical micro-CT parameters in male mice.

## Deletion of *Scarb1* in LysM-Cre-targeted cells does not affect body weight and fat mass

We then tested the alternative hypothesis that PC-OxPLs may exert their anti-osteogenic effects via activation of SCARB1 in macrophages, possibly leading to increased production of anti-osteoblastogenic cytokines, such as TNF-α [24]. To this end, we deleted *Scarb1* specifically in monocytes and macrophages by crossing *Scarb1*<sup>fl/fl</sup> mice with transgenic mice expressing the Cre recombinase under the control of LysM-Cre regulatory elements (LysM-Cre mice) [37]. The phenotype of the experimental *Scarb1*<sup>ΔLysM</sup> mice was compared with WT, LysM-Cre and *Scarb1*<sup>fl/fl</sup> littermate controls.

We quantified deletion of the *Scarb1* gene in bone marrow-derived macrophages from 6-month-old female mice. We found that the levels of *Scarb1* floxed exons were 88.7% lower in macrophages derived from *Scarb1*<sup>ΔLysM</sup> female mice as compared with LysM-Cre littermate controls, confirming deletion of the gene (S6 Fig). This level of deletion is comparable to the one obtained in other experiments in which we used LysM-Cre mice to delete other genes in the myeloid/osteoclast lineage [31,43,44]. There were no changes in *Scarb1* DNA levels in the spleen, confirming specificity of the deletion (S6 Fig).

In females, we did not observe any phenotypic differences between the four groups or differences in weight, fat mass, and lean mass at 4 and 6 months of age **(panels A-C in** S7 Fig**)**. In males, mice carrying the LysM Cre transgene (LysM-Cre and *Scarb1*<sup>ΔLysM</sup>) had lower weight than WT and *Scarb1*<sup>fl/fl</sup> mice (p = 0.021) and gained less weight with time (p = 0.003). Mice carrying the *Scarb1* floxed transgene (*Scarb1*<sup>fl/fl</sup> and *Scarb1*<sup>ΔLysM</sup>) had higher weight that WT and LysM-Cre (p = 0.015) but gained similar weight with time **(panel D in** S7 Fig **and** S3 Table**)**. *Scarb1*<sup>fl/fl</sup> mice alone had higher weight that the other three groups and gained more weight with time (p interaction 0.033) **(panel D in** S7 Fig **ad** S3 Table**)**. There was no difference in lean mass and fat mass between the four groups **(panels E-F in** S7 Fig**)**.

In summary, we could not detect in either sex and at any time point, any difference between Scarb1<sup>ΔLysM</sup> and the other three groups, indicating that, overall, deletion of *Scarb1* in myeloid progenitors does not affect weight, fat mass or lean mass.

## Deletion of *Scarb1* in myeloid cells does not affect bone mass

Bone mineral density (BMD) by DXA was measured at 4 and 6 months of age. In females, mice carrying the *Scarb1* floxed allele (*Scarb1*<sup>l/fl</sup> and *Scarb1*<sup>Δ ΔLysM</sup>) gained spinal BMD with time whereas WT and LysM-Cre mice lost BMD with time (p = 0.036); however, there were no differences in BMD at 6 months. (**Fig 4A** and S2 Table). No differences were found in femoral BMD between the four groups (**Fig 4B** and S2 Table). The analysis of total BMD indicated that WT mice lost BMD with time whereas the other three groups gained BMD with time (p interaction 0.014).

Micro-CT analysis, however, did not show any difference in vertebral and femoral cancellous bone and in the femoral cortical bone (Fig 4 D-4F; S8 Fig and panels A-C of S10 Fig).

In males, we did not observe any differences in spinal, femoral or total bone BMD between *Scarb1*<sup>ΔLysM</sup> mice and the three groups of littermate controls (WT, LysM-Cre and *Scarb1*<sup>fl/fl</sup>) at any time point (Fig 5A-5C**).** Similarly, micro-CT analysis of 6-month-old male mice did not show any differences in vertebral cancellous bone and in femoral cancellous and cortical bone between *Scarb1*<sup>ΔLysM</sup> mice and the controls (Fig 5D-5F, S9 Fig **and panels D-F of** S10 Fig).

Overall, these results indicate that deletion of *Scarb1* in cells of the myeloid lineage does not alter bone mass.

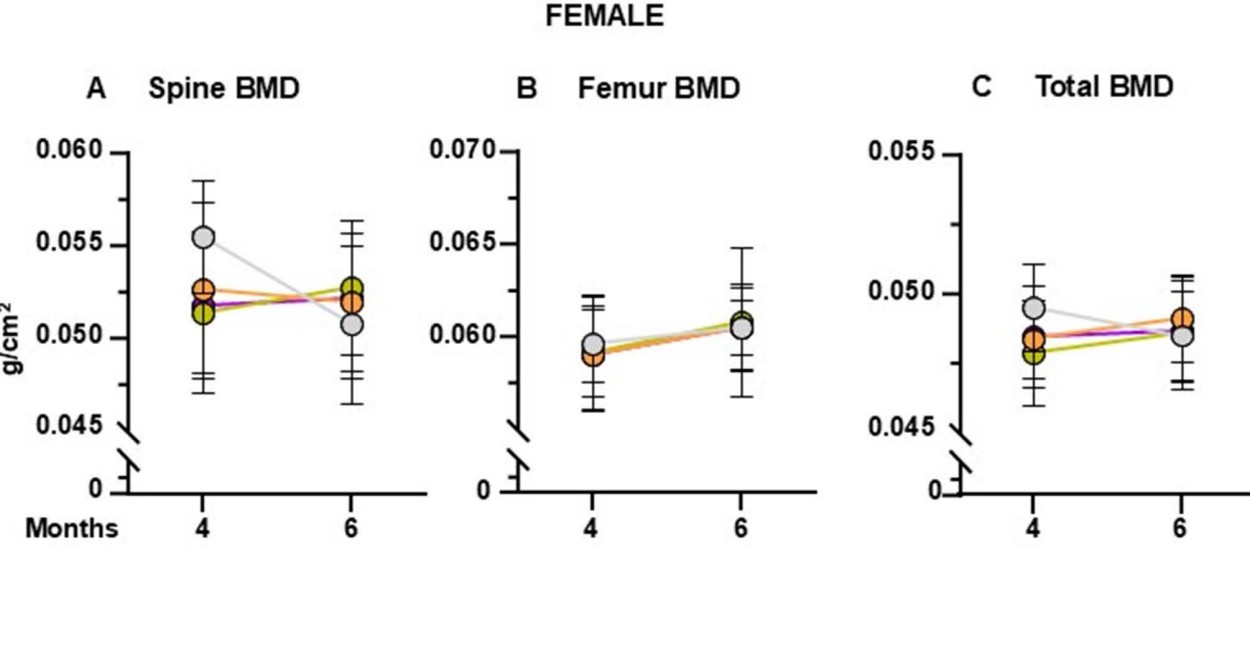

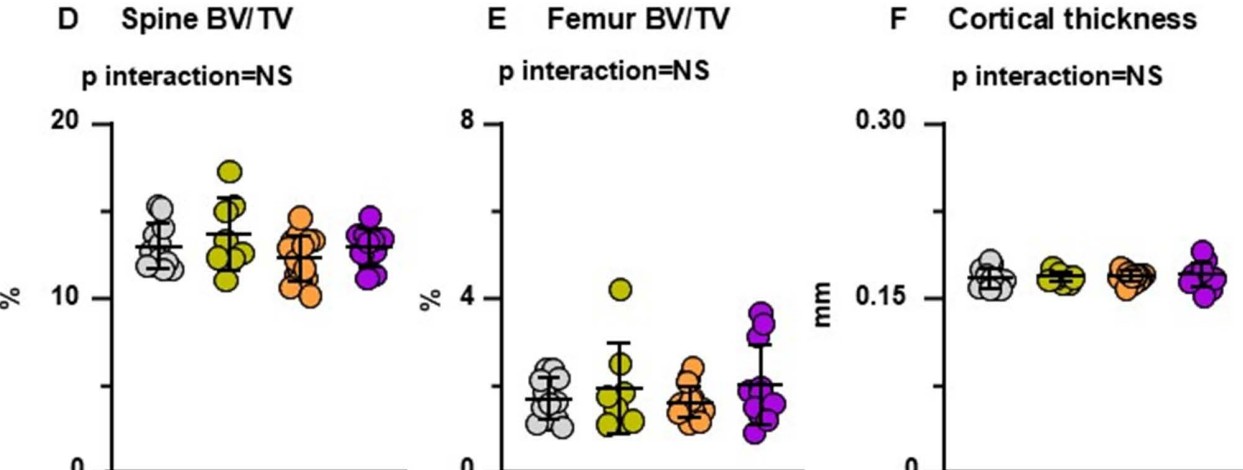

**Fig 4. Deletion of *Scarb1* in LysM-Cre expressing cells does not affect bone mass in females.** (A-C) Determinations of spinal, femoral and total BMD by DXA in 4- and 6-month-old mice [WT n = 12; *Scarb1*fl/fl n = 8; LysM-Cre n = 15; *Scarb1*ΔLysM n = 12]. Data are shown as mean and standard deviation. Adjusted p-values <0.05, calculated by repeated measures using two-way ANOVA, are shown in S2 Table. BMD: bone mineral density. (D-F) Micro-CT analysis of cancellous bone in vertebra and femoral metaphysis, and cortical bone at midshaft in 6-month-old mice [WT n = 12; *Scarb1*fl/fl n = 8; LysM-Cre n = 15; *Scarb1*ΔLysM n = 12]. Data analyzed by 2-way ANOVA; the p-values were adjusted using the Tukey's pairwise comparison procedure. BV/TV, bone volume/total volume. NS: not significant, p > 0.05.

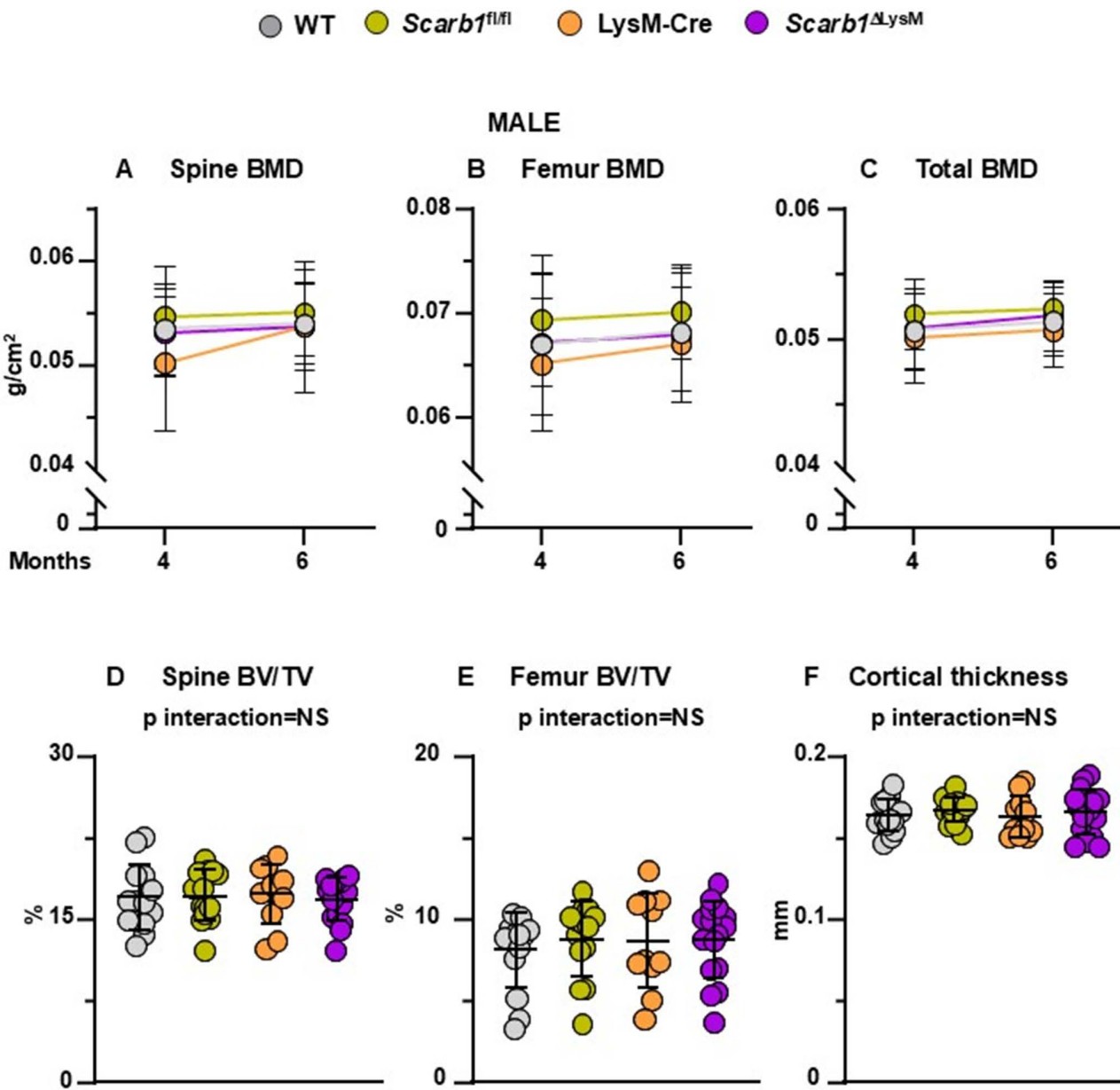

**Fig 5. Deletion of *Scarb1* in LysM-Cre expressing cells does not affect bone mass in males.** (A-C) Determinations of spinal, femoral and total BMD by DXA in 4- and 6-month-old mice [WT n = 14; *Scarb1*fl/fl n = 14; LysM-Cre n = 11; *Scarb1*ΔLysM n = 16]. Data are shown as mean and standard deviation.. BMD: bone mineral density. (D-F) Micro-CT analysis of cancellous bone in vertebra and femoral metaphysis, and cortical bone at midshaft in 6-month-old males [WT n = 14; *Scarb1*fl/fl n = 14; LysM-Cre n = 11; *Scarb1*ΔLysM n = 16]. Data analyzed by 2-way ANOVA; the p-values were adjusted using the Tukey's pairwise comparison procedure. BV/TV, bone volume/total volume. NS: not significant, p > 0.05.

## Discussion

The results presented herein show that deletion of *Scarb1* in either the entire osteoblast lineage or in the myeloid/osteoclast lineage does not affect bone mass in either female or male mice indicating that *Scarb1* expression in osteoblasts or osteoclasts does not play a major role in skeletal homeostasis.

Previous reports implicated SCARB1 in bone metabolism. *In vitro* studies showed that, in osteoblasts, *Scarb1* is responsible for selective uptake of cholesteryl esters and estradiol from HDL and LDL [19]. The uptake of these lipoproteins however was similar in osteoblastic cells from WT and *Scarb1* KO mice, indicating that this process was not dependent on SCARB1 in these cells [19,23]. Osteoblastic cells derived from *Scarb1* KO mice exhibit increased proliferation, increased alkaline phosphatase activity, enhanced matrix mineralization, and higher expression of the osteoblastogenic transcription factors Sp7 and Runx2 [19,23]. Silencing *Scarb1* or the absence of *Scarb1* in osteoblastic cells attenuates both osteoblast apoptosis and the decrease in differentiation of osteogenic precursors induced by OxLDL (9). Moreover, we show here that the absence of *Scarb1* prevents suppression of proliferation by OxLDL. On the other hand, others have reported that *Scarb1* is indispensable for HDL-induced proliferation of rat mesenchymal stem cells [45].

Earlier attempts to determine the bone phenotype of mice with global deletion of *Scarb1* (*Scarb1* KO) have produced conflicting results. Martineau and colleagues reported that *Scarb1* KO mice have increased cancellous bone at 2 and 4 months of age and this increase was associated with increased osteoblast surface, mineralized surface, and bone formation rate with no changes in osteoclasts parameters [19,22,23]. In contrast, Tourkova et al. showed that *Scarb1* KO mice have low bone mass at 16 weeks of age with low bone formation and decreased osteoclastogenesis compared to WT mice, suggesting that SCARB1 is required for osteoblast and osteoclast differentiation [46]. All those studies, however, were performed in global knockout mice.

The purpose of our study was to investigate the role of SCARB1 specifically in osteoblasts as a first step in determining whether it mediates the effects of PC-OxPL on this cell type. SCARB1 is the most abundant scavenger receptor in osteoblasts that is known to bind PC-OxPL, which has deleterious effects on bone homeostasis mainly by affecting osteoblasts [9,14]. Male and female mice expressing an antibody fragment (E06-scFv) that neutralizes PC-OxPL, exhibit increased cancellous and cortical bone mass at 6 months of age [9] and are protected from the deleterious effect of aging on bone [18]. E06-scFV increases osteoblast number and activity and decreases osteoblast apoptosis, indicating that PC-OxPL affects osteoblasts under physiological conditions [9]. The results presented herein, however, clearly demonstrate that expression of *Scarb1* in osteoblasts is not required for bone mass acquisition and therefore is not an essential mediator of the deleterious effects of PC-OxPL in osteoblasts.

Therefore we tested the alternative possibility that PC-OxPLs may exert their anti-osteogenic effects via activation of SCARB1 in macrophages [24].

The role of SCARB1 in macrophages has been extensively studied in mouse models of atherosclerosis where it has been found to be both pro-atherogenic and anti-atherogenic [47]. Whereas earlier reports indicate that deletion of *Scarb1* reduced the development of atherosclerosis [48], another study showed that deletion of this receptor in monocytes and macrophages worsened the extension of atherosclerotic lesions at earlier stages [49]. Specifically, *Scarb1* deficiency in LysM-Cre expressing cells increased atherosclerosis by increasing the expression of the apoptosis inhibitor of macrophages (AIM) protein and consequently reducing macrophage apoptosis. Since macrophage apoptosis is associated with attenuation of early atherogenesis, this study suggests that decreased apoptosis is responsible for the increased accumulation of macrophages in the atherosclerotic plaque and expansion of the lesions. In more advanced atherosclerotic lesions SCARB1 present in macrophages binds and mediates the removal of apoptotic cells by efferocytosis; the absence of *Scarb1*, therefore, increases the numbers of apoptotic cells that did not get removed and increases the necrosis of the atherosclerotic plaques [50]. Moreover, *Scarb1* expression in macrophages induces expression of transcription factor EB (TFEB), a master regulator of autophagy, which limits the necrosis and increases stability of atherosclerotic plaques; deletion of *Scarb1*, therefore, impairs autophagy and worsens the atherosclerosis at later stages [51]. Heretofore, the role of SCARB1 in macrophages has not been studied in bone.

The results presented herein show that deletion of *Scarb1* in cells of the myeloid/osteoclast lineage does not affect bone mass in either female or male mice. This evidence indicates that *Scarb1* expression in macrophages and

osteoclasts does not play a major role in skeletal homeostasis. Therefore, SCARB1 in these cells is not a major mediator of the deleterious effects of PC-OxPLs on bone.

PC-OxPLs bind to other scavenger receptors, such as CD36, and toll-like receptors, such as TLR2, 4 and 6 which may mediate the deleterious effects in bone and compensate for the lack of SCARB1 [14]. Our previous work has indicated that PC-OxPLs decreases Wnt signaling, and this decrease may mediate the negative effects of PC-OxPLs in bone [18]. Thus, identification of the mechanisms by which PC-OxPLs affect bone homeostasis will require further investigation.

An important limitation of this study is that we did not challenge the mice with high fat diet or high cholesterol diet. It remains possible that SCARB1 in myeloid progenitors may play a role in inflammatory conditions with higher levels of oxidized phospholipids. This possibility will be pursued in future studies.

Finally, we acknowledge that the results presented herein do not explain the bone phenotype of the *Scarb1* KO mice. Our results suggest that the effect of *Scarb1* deletion on bone homeostasis is unlikely to be related to the expression in osteoblasts or osteoclasts alone. It is possible that the effects of Scarb1 are complex and mediated by multiple cell types.

## Supporting information

**S1 Fig. *Scarb1* gene was effectively deleted in bone.** Quantitative PCR (qPCR) of genomic DNA isolated from femoral and tibial cortical bone and spleen in 6-month-old females [Osx1-Cre n = 4, Osx1-Cre; *Scarb1*$^{\Delta OSX1}$n=4] (A) and in 6-month-old males [Osx1-Cre n = 4, *Scarb1*$^{\Delta OSX1}$n=4] (B). Data are shown as mean and standard deviation. Data analyzed by unpaired T-test.
(TIF)

**S2 Fig. Deletion of *Scarb1* in Osx1-Cre expressing cells does not affect weight, fat or lean mass in both sexes.** Measurements of weight, fat mass and lean mass in 2- and 6-month-old females [WT n = 17; *Scarb1*$^{fl/fl}$ n = 9; Osx1-Cre n = 14; *Scarb1*$^{\Delta OSX1}$n=10] (A-C) and 2- and 6-month-old males [WT n = 11; *Scarb1*$^{fl/fl}$ n = 14; Osx1-Cre n = 12; *Scarb1*$^{\Delta OSX1}$n=12] (D-F). Data are shown as mean and standard deviation. Adjusted p-values <0.05, calculated by repeated measures using two-way ANOVA, are shown in S3 Table.
(TIF)

**S3 Fig. Deletion of *Scarb1* in Osx1-Cre expressing cells does not affect microarchitecture of cancellous bone in female mice.** Micro-CT analysis of cancellous bone architecture in 6-month-old females. (A) Trabecular number, (B) trabecular thickness, and (C) trabecular separation of vertebral cancellous bone. (D) Trabecular number, (E) trabecular thickness, and (F) trabecular separation of femoral cancellous bone [WT n = 17; *Scarb1*$^{fl/fl}$ n = 9; Osx1-Cre n = 13; *Scarb1*$^{\Delta OSX1}$n=10]. Data are shown as mean and standard deviation. Data analyzed by 2-way ANOVA; the p-values were adjusted using the Tukey's pairwise comparison procedure. Tb, trabecular.
(TIF)

**S4 Fig. Deletion of *Scarb1* in Osx1-Cre expressing cells does not affect microarchitecture of cancellous bone in male mice.** Micro-CT analysis of cancellous bone architecture in 6-month-old males. (A) Trabecular number, (B) trabecular thickness, and (C) trabecular separation of vertebral cancellous bone. (D) Trabecular number, (E) trabecular thickness, and (F) trabecular separation of femoral cancellous bone [WT n = 11; *Scarb1*$^{fl/fl}$ n = 14; Osx1-Cre n = 12; *Scarb1*$^{\Delta OSX1}$n=12]. Data are shown as mean and standard deviation. Data analyzed by 2-way ANOVA; the p-values were adjusted using the Tukey's pairwise comparison procedure. Tb, trabecular.
(TIF)

**S5 Fig. Deletion of *Scarb1* using Osx1-Cre does not affect cortical bone in female or femoral length in female or male mice.** Micro-CT analysis of cortical bone architecture and femoral length. (A) Total area, (B) Medullary area, (C)

Femoral length in 6-month-old females [WT n = 17; *Scarb1*$^{fl/fl}$ n = 9; Osx1-Cre n = 13; *Scarb1*$^{\Delta OSX1}$n=10]. (D) Total area, (E) Medullary area, (F) Femoral length in 6-month-old males [total area and medullary area WT n = 11; *Scarb1*$^{fl/fl}$ n = 14; Osx1-Cre n = 12; *Scarb1*$^{\Delta OSX1}$n=12], [femoral length WT n = 11; *Scarb1*$^{fl/fl}$ n = 13, Osx1-Cre n = 12; *Scarb1*$^{\Delta OSX1}$n=12]. Data are shown as mean and standard deviation. Data analyzed by 2-way ANOVA; the p-values were adjusted using the Tukey's pairwise comparison procedure.
(TIF)

**S6 Fig.  *Scarb1* gene was effectively deleted in bone marrow macrophages.** Quantitative PCR (qPCR) of genomic DNA isolated from bone marrow-derived macrophages and spleen of 6-month-old female mice [LysM-Cre n = 3, *Scarb1*$^{\Delta LysM}$ n = 3]. Data are shown as mean and standard deviation. Data analyzed by unpaired T-test.
(TIF)

**S7 Fig.  Deletion of *Scarb1* in LysM-Cre expressing cells does not affect weight, fat or lean mass in both sexes.** Measurements of weight, fat mass and lean mass in 4- and 6-month-old females [WT n = 12; *Scarb1*$^{fl/fl}$ n = 8; LysM-Cre n = 15; *Scarb1*$^{\Delta LysM}$ n = 12] (A-C) and 4- and 6-month-old males [WT n = 14; *Scarb1*$^{fl/fl}$ n = 14; LysM-Cre n = 11; *Scarb1*$^{\Delta LysM}$ n = 16]. Data are shown as mean and standard deviation. Adjusted p-values <0.05, calculated by repeated measures using two-way ANOVA, are shown in S3 Table.
(TIF)

**S8 Fig.  Deletion of *Scarb1* in LysM-Cre expressing cells does not affect microarchitecture of cancellous bone in female mice.** Micro-CT analysis of cancellous bone architecture in 6-month-old females. (A) Trabecular number, (B) trabecular thickness, and (C) trabecular separation of vertebral cancellous bone. (D) Trabecular number, (E) trabecular thickness, and (F) trabecular separation of femoral cancellous bone [WT n = 12; *Scarb1*$^{fl/fl}$ n = 8; LysM-Cre n = 15; *Scarb1*$^{\Delta LysM}$ n = 12]. Data are shown as mean and standard deviation. Data analyzed by 2-way ANOVA; the p-values were adjusted using the Tukey's pairwise comparison procedure. Tb, trabecular.
(TIF)

**S9 Fig.  Deletion of *Scarb1* in LysM-Cre expressing cells does not affect microarchitecture of cancellous bone in male mice.** Micro-CT analysis of cancellous bone architecture in 6-month-old males. (A) Trabecular number, (B) trabecular thickness, and (C) trabecular separation of vertebral cancellous bone. (D) Trabecular number, (E) trabecular thickness, and (F) trabecular separation of femoral cancellous bone [WT n = 14; *Scarb1*$^{fl/fl}$ n = 14; LysM-Cre n = 11; *Scarb1*$^{\Delta LysM}$ n = 16]. Data are shown as mean and standard deviation. Data analyzed by 2-way ANOVA; the p-values were adjusted using the Tukey's pairwise comparison procedure. Tb, trabecular.
(TIF)

**S10 Fig.  Deletion of *Scarb1* using LysM-Cre does not affect cortical bone in female or femoral length in female or male mice.** Micro-CT analysis of cortical bone architecture and femoral length. (A) Total area, (B) Medullary area, (C) Femoral length in 6-month-old females [WT n = 12; *Scarb1*$^{fl/fl}$ n = 8; LysM-Cre n = 15; LysM-Cre; *Scarb1*$^{fl/fl}$ n = 12], (D) Total area, (E) Medullary area, (F) Femoral length in 6-month-old males [WT n = 14; *Scarb1*$^{fl/fl}$ n = 14; LysM-Cre n = 11; *Scarb1*$^{\Delta LysM}$ n = 16]. Data are shown as mean and standard deviation. Data analyzed by 2-way ANOVA; the p-values were adjusted using the Tukey's pairwise comparison procedure.
(TIF)

**S1 Table.  TaqMan assays.** List of the TaqMan primers used for quantification of mRNA and genomic DNA by qPCR.
(DOCX)

**S2 Table.  List of adjusted p values <0.05 for the indicated figures.**
(DOCX)

**S3 Table. List of adjusted p values <0.05 for the indicated figures.**
(DOCX)

**S1 File. Raw data for all the Figures and Supporting information present in the manuscript.**
(XLSX)

## Acknowledgments

We thank Stuart B Berryhill for technical assistance. This study is the results of work supported with resources and the use of facilities at the University of Arkansas for Medical Sciences and the Central Arkansas Veterans Healthcare System, Little Rock, AR. The contents do not represent the views of the U.S. Department of Veterans Affairs or the U.S. Government.

## Author contributions

**Conceptualization:** Stavros C Manolagas, Charles A O'Brien, Elena Ambrogini.

**Data curation:** Michela Palmieri, Teenamol E Joseph, Horacio Gomez-Acevedo, Ha-neui Kim, Charles A O'Brien, Elena Ambrogini.

**Formal analysis:** Horacio Gomez-Acevedo, Charles A O'Brien, Elena Ambrogini.

**Funding acquisition:** Charles A O'Brien, Elena Ambrogini.

**Investigation:** Charles A O'Brien, Elena Ambrogini.

**Methodology:** Ha-neui Kim, Charles A O'Brien, Elena Ambrogini.

**Project administration:** Michela Palmieri, Teenamol E Joseph, Elena Ambrogini.

**Supervision:** Charles A O'Brien, Elena Ambrogini.

**Validation:** Charles A O'Brien, Elena Ambrogini.

**Writing – original draft:** Elena Ambrogini.

**Writing – review & editing:** Michela Palmieri, Teenamol E Joseph, Horacio Gomez-Acevedo, Ha-neui Kim, Stavros C Manolagas, Charles A O'Brien, Elena Ambrogini.

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
