## [Decision Letter · Decision Letter 0]

6 Sep 2025

Dear Dr. Ambrogini,

Thank you for submitting your manuscript to PLOS ONE. We apologize for the lengthy review timeline. After careful consideration, we feel that it has merit but does not fully meet PLOS ONE’s publication criteria as it currently stands. Therefore, we invite you to submit a revised version of the manuscript that addresses the points raised during the review process.

We look forward to receiving your revised manuscript.

Kind regards,

Joseph L Roberts, PhD

Academic Editor

PLOS ONE

Journal Requirements:

4. Please upload a copy of Supporting Information Table 1 which you refer to in your text on page 10.

Reviewers' comments:

Reviewer's Responses to Questions

**Comments to the Author**

1. Is the manuscript technically sound, and do the data support the conclusions?

Reviewer #1: Yes

2. Has the statistical analysis been performed appropriately and rigorously?

Reviewer #1: I Don't Know

3. Have the authors made all data underlying the findings in their manuscript fully available?

Reviewer #1: Yes

4. Is the manuscript presented in an intelligible fashion and written in standard English?

Reviewer #1: Yes

Reviewer #1: In this interesting study, the authors examined the effects of targeted deletion of Scarb1 in osteoblasts and myeloid cells on bone mass in male and female mice. Using Osx-Cre:Scarb1fl/fl and LysM-Cre:Scarb1fl/fl conditional knockout (cKO), they report no significant differences in trabecular or cortical bone mass in the femur or spine compared to controls. Based on these findings, the authors conclude that Scarb1 in either the osteoblast or myeloid cell lineage is not critical for bone homeostasis.

This study addresses an important and novel question by attempting to identify the specific cell types through which the scavenger receptor Scarb1 mediates the toxic effects of oxidized phospholipids (OxPLs) in bone. The use of male and female mice is valuable because it addresses the possibility of sex dimorphism. However, the study is largely descriptive, relying primarily on DXA and microCT data to demonstrate that such effects are not mediated by Scarb1 expressed in osteoblast progenitors or myeloid cells, without giving any further functional insights into possible underlying mechanisms.

Overall, the manuscript is well-written and presents findings that are relevant to the broader scientific bone research community. However, several important concerns must be addressed before the manuscript can be considered suitable for publication.

Major

-OxPLs are typically associated with diseased or inflammatory states. Yet, the authors conducted their experiments on relatively young (6-month-old) animals without challenging them with a high-fat or high-cholesterol diet. This represents an important limitation of the study, making it difficult to assess the relevance of the findings to pathological settings. The authors briefly acknowledge this limitation in the Discussion (lines 520-523). The rationale for studying unchallenged animals should be clarified in the Introduction to help readers who are less familiar with the OxPLs field to better understand the study design.

-In the first experiment (Figure 1), the authors compare calvaria-derived osteoblasts from WT and Scarb1 global KO. This experiment feels somewhat disconnected from the rest of the manuscript, which centers on conditional deletion of Scarb1 in specific bone cells. The rationale behind this experiment is not clearly stated, and the connection to the subsequent cKO studies is underdeveloped. In particular, the authors report significant differences in osteoblast differentiation and proliferation between WT and Scarb1 KO, but do not follow up with similar studies using cells from the Osx1-Cre or LysM-Cre cKO models. This represents a missed opportunity to validate whether these cell-specific deletions replicate the phenotype seen in the global KO.

Figure 1A:

1. Osteocalcin and ALP levels appear unusually low for 21-day osteoblast cultures, especially for the WT. The text in Line 220 mentions the use of ΔCt for quantification but refers to a different experiment. The authors should clarify whether ΔCt or ΔΔCt was used in this experiment. If ΔCt was used, data should be re-analyzed using the standard ΔΔCt method, or the decision to use ΔCt should be justified. Additionally, there is no detailed description of RNA isolation, cDNA synthesis, or qPCR in the Methods section. A dedicated paragraph should be added. Justification for using MRPS2 as a housekeeping gene should also be included, as this is not a commonly used reference gene.

2. The authors should have investigated mineralization and ALP activity in the same cell cultures to validate the biological relevance of gene expression data.

Figure 1B:

1. A representative figure should be included alongside the quantification to support the proliferation data.

Minor

-Bone mass is dependent on a balance between bone formation and bone resorption. Even though the overall bone mass is unchanged in both cKO mouse lines, processes of bone formation and resorption could still be altered. However, the authors do not present any data assessing osteoblast or osteoclast differentiation or function. Functional analyses such as bone histomorphometry or in vitro assays using calvaria-derived osteoblasts and bone marrow-derived osteoclasts would strengthen the manuscript and provide a deeper understanding of the cellular consequences of specific Scarb1 deletion.

-Osx1-Cre is known to be aspecific, targeting not only osteoblasts, osteocytes, and hypertrophic chondrocytes, but also bone marrow stromal cells, adipocytes, and perivascular cells. Notably, this Cre line also affects cells outside the skeletal system, including olfactory glomerular cells and subsets of gastric and intestinal epithelium (Chen et al., PLoS One 2014). Given this broad expression pattern, it raises the question: why was Osx1-Cre selected instead of more specific Cre drivers such as Runx2-Cre or Col1a1-Cre, which more selectively target osteoblast-lineage cells?

Moreover, Osx1-Cre alone appears to induce a phenotype, as Osx1-Cre controls differ significantly from other control groups in most experiments (Figure 2, Supplementary Figure 3, Supplementary Figure 4, Supplementary Figure 5). While it is understandable that establishing and characterizing another Cre line would be resource- and time-consuming, the authors should explicitly acknowledge the limitations of the Osx1-Cre driver and provide a clearer rationale for its use in both the Methods and Discussion sections.

-The use of three separate control groups (WT, Osx1-Cre, and Scarb1fl/fl), while useful to dissect the effects of the Cre and the loxP sites alone, is uncommon and creates some conflicts in the interpretation of the analyses. In some of the μCT results, for example in Supplementary Figures 3D-F and 4D-F, WT and Scarb1fl/fl controls differ significantly from the cKO, but the authors logically dismiss these differences based on a lack of significance relative to the Osx1-Cre control. If Osx1-Cre introduces changes independent of Scarb1 deletion, then it may be more appropriate to use Osx1-Cre as the sole control. As a suggestion, the comparisons to Osx1-Cre could be shown in the main paper, and the current figures might be moved to the Supplementary. Alternatively, the authors should clearly articulate the reasons for including each control.

Similarly, three control groups were used for the LysM-Cre experiment. Differently than Osx1-Cre, LysM-Cre does not result in a skeletal phenotype alone (Dallas et al., Curr Osteoporos Rep. 2018). As discussed above, a justification for the use of multiple control groups should be provided, or the controls should be streamlined to those that are biologically and statistically appropriate.

-In the Abstract, the authors mention that skeletal analysis of Scarb1 KO mice produced contradictory results (lines 33-35). Since the current data indicate that neither the osteoblast nor the myeloid/osteoclast lineage mediates Scarb1 role in bone homeostasis, it would be helpful if the authors could comment on this in the Discussion. Do they have any ideas about what might explain the discrepancies in the earlier findings, or how their results fit into that context?

Line 271: sample number reported in Figure 1 caption “C57BL6/J (n=5) and Scarb1 KO mice (n=2)” does not match what is shown in the figure (WT n=3, KO n=3).

Line 274: “4-5-month-old WT and Scarb1 KO mice (n=3/group)”, again numbers do not match with what is shown in the graph.

Line 419: “Osx1-Cre mice lost BMD with time […] Figure 4A”. The authors were probably referring to LysM-Cre, not to Osx1-Cre.

Supplementary Figures 2A-B-C, 3A-B-C, 4A-B-C, 5A-B-C: p values are missing from the Figures and the figure legend.

**Do you want your identity to be public for this peer review?** For information about this choice, including consent withdrawal, please see our Privacy Policy

Reviewer #1: No

---

## [Author Response · Author response to Decision Letter 1]

27 Sep 2025

please see the document attached

---

## [Decision Letter · Decision Letter 1]

20 Oct 2025

Deletion of the scavenger receptor Scarb1 in osteoblast progenitors and myeloid cells does not affect bone mass

PONE-D-25-36929R1

Dear Dr. Ambrogini,

We’re pleased to inform you that your manuscript has been judged scientifically suitable for publication and will be formally accepted for publication once it meets all outstanding technical requirements.

Kind regards,

Joseph L Roberts, PhD

Academic Editor

PLOS ONE

Additional Editor Comments (optional):

Reviewers' comments:

Reviewer's Responses to Questions

**Comments to the Author**

Reviewer #1: All comments have been addressed

2. Is the manuscript technically sound, and do the data support the conclusions?

Reviewer #1: Yes

3. Has the statistical analysis been performed appropriately and rigorously?

Reviewer #1: Yes

4. Have the authors made all data underlying the findings in their manuscript fully available?

Reviewer #1: Yes

5. Is the manuscript presented in an intelligible fashion and written in standard English?

Reviewer #1: Yes

Reviewer #1: The authors addressed my comments. They state they will make all the data underlying the findings fully available through public repository at the time of publication. I do not have any other comments.

**Do you want your identity to be public for this peer review?** For information about this choice, including consent withdrawal, please see our Privacy Policy

Reviewer #1: No

---

## [Editor Report · Acceptance letter]

PONE-D-25-36929R1

PLOS ONE

Dear Dr. Ambrogini,

I'm pleased to inform you that your manuscript has been deemed suitable for publication in PLOS ONE. Congratulations! Your manuscript is now being handed over to our production team.

Kind regards,

on behalf of

Dr. Joseph L Roberts

Academic Editor

PLOS ONE